# Oxidative Stress Implication in Retinal Diseases—A Review

**DOI:** 10.3390/antiox11091790

**Published:** 2022-09-10

**Authors:** Marcella Nebbioso, Federica Franzone, Alessandro Lambiase, Vincenza Bonfiglio, Paolo Giuseppe Limoli, Marco Artico, Samanta Taurone, Enzo Maria Vingolo, Antonio Greco, Antonella Polimeni

**Affiliations:** 1Department of Sense Organs, Faculty of Medicine and Odontology, Sapienza University of Rome, p.le A. Moro 5, 00185 Rome, Italy; 2Turin Hospital Ophthalmic, via F. Juvarra 19, 10122 Turin, Italy; 3Department of Experimental Biomedicine and Clinical Neuroscience, Ophthalmology Section, University of Palermo, Via del Vespro 129, 90127 Palermo, Italy; 4Low Vision Research Centre of Milan, p.zza Sempione 3, 20145 Milan, Italy; 5IRCCS, Fondazione Bietti, via Livenza 3, 00198 Rome, Italy; 6Department of Oral and Maxillofacial Sciences, Sapienza University of Rome 5, p.le A. Moro 5, 00185 Rome, Italy

**Keywords:** age-related macular degeneration, diabetic retinopathy, Eales’ disease, macular diseases, neurodegeneration, oxidative stress, reactive oxygen species, nitrosative stress, retinitis pigmentosa, retinopathy of prematurity, Stargardt disease

## Abstract

Oxidative stress (OS) refers to an imbalance between free radicals (FRs), namely highly reactive molecules normally generated in our body by several pathways, and intrinsic antioxidant capacity. When FR levels overwhelm intrinsic antioxidant defenses, OS occurs, inducing a series of downstream chemical reactions. Both reactive oxygen species (ROS) and reactive nitrogen species (RNS) are produced by numerous chemical reactions that take place in tissues and organs and are then eliminated by antioxidant molecules. In particular, the scientific literature focuses more on ROS participation in the pathogenesis of diseases than on the role played by RNS. By its very nature, the eye is highly exposed to ultraviolet radiation (UVR), which is directly responsible for increased OS. In this review, we aimed to focus on the retinal damage caused by ROS/RNS and the related retinal pathologies. A deeper understanding of the role of oxidative and nitrosative stress in retinal damage is needed in order to develop targeted therapeutic interventions to slow these pathologies.

## 1. Introduction

Since the term “oxidative stress” (OS) was introduced 30 years ago [1], this concept has been extensively studied in various fields of medicine, including ophthalmology. Specifically, OS refers to an imbalance between free radicals (FRs), namely highly reactive molecules normally generated in our body by several pathways, and intrinsic antioxidant capacity [2]. Until the mid-1970s, the only term used was “FRs”: subsequently, oxygen and nitrogen radicals were distinguished with the introduction of “reactive oxygen species” (ROS) and “reactive nitrogen species” (RNS) [3].

Nowadays, the scientific literature focuses more on the involvement of ROS in the pathogenesis of anterior and posterior segment eye diseases than on the role played by RNS. Both ROS and RNS are produced by numerous chemical reactions that take place in tissues and organs and are then eliminated by antioxidant molecules. When FR levels overwhelm intrinsic antioxidant defenses, OS occurs, inducing a series of downstream chemical reactions [4]. By its very nature, the eye is highly exposed to ultraviolet radiation (UVR), which is directly responsible for increased OS. This is especially true in children due to the transparency of ocular media [5]. Both anterior and posterior eye structures are a possible target of OS, and several pathologies are related to increased FRs, such as dry eye syndrome, keratitis, cataracts, glaucoma [6], age-related macular degeneration [4], diabetic retinopathy (DR) and others [7].

In this review, we wanted to focus our knowledge on the retinal damage caused by ROS and RNS on the related retinal and macular pathologies. Therefore, we believe a deeper understanding of the role of OS in retinal damage is needed in order to favor the development of therapeutic interventions aimed at blocking or at least slowing the ocular pathologies of the posterior segment.

## 2. Reactive Oxygen Species (ROS) and Reactive Nitrogen Species (RNS)

Current evidence suggests that low levels of ROS are physiologically formed in the human body as key molecules in the regulation of several cellular functions such as gene expression, transcription factor synthesis, inflammation, apoptosis and wound healing; they play a crucial role in maintaining cell homoeostasis [8,9]. This can be supported with two examples. The first is apoptosis, which is an indispensable process for normal tissue homeostasis, as demonstrated by all the diseases related to aberrations in the apoptosis pathway. The second is the role of ROS in maintaining the eye’s innate immunity. It is not surprising that Müller glial cells (MGC) in the retina are able to produce abundant quantities of anti-bacterial ROS [9].

In the case of excessive ROS generation or a reduction in antioxidant defenses, diseases may develop as a consequence of a shift from redox to a more oxidizing state of a specific biological system [7]. ROS are the result of a partial reduction of molecular oxygen and comprise hydrogen peroxide (H_2_O_2_), superoxide anion (O_2_^●−^), hydroxyl radical (·OH), peroxyl radical (ROO) and singlet oxygen (O_2_) [7].

Both exogenous and endogenous sources of ROS have been identified. The former include UVR, environmental pollution and tobacco smoke and are a significant source of oxidative products, while the latter involve especially the mitochondria, namely the site of the respiratory chain: superoxide radical generation takes place via the single-electron leak at respiratory complexes I and III of the oxidative phosphorylation (OXPHOS) pathway [8].

In addition, another important endogenous source of ROS is represented by the family of nicotinamide adenine dinucleotide phosphate (NADPH) oxidases (Nox): Nox1, Nox2, Nox3, Nox4, Nox5, dual oxidase 1 (Duox1) and dual oxidase 2 (Duox2), which are enzymes of the same class that provide for either superoxide or hydrogen peroxide formation, depending on the isoform. As regards the superoxide anion (O_2_^●−^), it is formed by oxygen reduction and subsequently turns into hydrogen peroxide, a low reactive molecule that is responsible for the formation of the hydroxyl radical, the most reactive ROS, via Fenton’s reaction [8].

As already said for ROS, RNS also are normally produced in the human body and, more specifically, in the eye. Nitric oxide (NO) is a molecule characterized by an extremely short half-life that is able to diffuse across cell membranes [10]. A normal NO concentration in the retina is crucial for vision processes, but an excess is extremely dangerous. The enzyme involved in NO synthesis is nitric oxide synthase (NOS), which catalyzes the chemical reaction in the presence of oxygen and NADPH. There are three NOS isoforms: nNOS (NOS-I), which represents the constitutive calcium-dependent neuronal NOS and is typically expressed in neurons and skeletal muscles; eNOS (NOS II), which is the endothelial NOS isoform and is, therefore, usually found in vascular endothelial cells (ECs); and iNOS (NOS III), also known as the inducible or calcium-independent NOS isoform, which is induced in case of pathological stress [11,12]. At the level of the retina, NO plays several different functions, such as modulating the blood flow and maintaining normal visual function by producing guanosine 3′, 5′-cyclic monophosphate (cGMP), which is essential in the light transduction process of photoreceptors (PRs) [11,12].

Just as several oxygen and nitrogen radicals exist, there are also different antioxidant defenses in the human body: small molecular weight and non-enzymatic antioxidants such as vitamins A, C and E, and intracellular antioxidant enzymes such as catalase, glutathione peroxidase (GPx), glutathione reductase (GR), copper/zinc-superoxide dismutase (Cu/Zn-SOD, SOD1) in cytosol, manganese superoxide dismutase (Mn-SOD) in the mitochondrial matrix as well as carotenoids, α-lipoic acid and minerals such as Cu and Zn [8]. As a consequence of excessive ROS and RNS, serious damage to proteins, lipids and DNA occurs. For instance, in the latter case, when ROS attack the sugar phosphate backbone and the nitrogenous bases, DNA breaks occur. Mitochondrial DNA (mtDNA) seems to be more prone to oxidative damage than nuclear DNA (nDNA). This is probably explained by the absence of an efficient DNA repair system. This condition triggers mtDNA mutations with a series of alterations in the respiratory chain as well as a certain amount of ROS production.

Nowadays, numerous markers of OS are known, including 4-hydroxynonenal and malondialdehyde, a marker of lipid peroxidation; 8-hydroxy-2-deoxyguanosine, a marker of DNA damage; and protein carbonyl groups, markers of protein oxidation [8].

## 3. Macula, Retina and Oxidative Stress (OS)

The human retina is a highly energy-consuming sensory tissue formed by several cell layers derived from the neuroectoderm. It is organized into two different structures: the neuroretina and retinal pigment epithelium (RPE). At the posterior pole of the eye, there is a yellow region measuring about 5 to 6 mm in diameter called the “macula”; the term refers to the presence of a series of xanthophyll pigments, lutein and zeaxanthin, which are responsible for the color of this region (macula lutea). The macula ensures high acuity vision, which is due to the high concentration of cones here [13].

Considering its high metabolism, the macular region is constantly exposed to high levels of ROS. These molecules are involved in a number of cellular reactions that are essential for physiological processes. The main sources of ROS are cellular enzymes such as cytochrome p450 enzymes, cyclooxygenases, lipoxygenases, xanthine oxidase, mitochondrial complex I and III cytochrome chains and others. In addition, due to the high blood flows in the macula, abundant exogenous ROS are present in this area [14].

Numerous cells are observed in the retina: PRs, glial cells, interneurons and projection neurons. From the inside to the outside, nine layers can be recognized in the neuroretina: internal limiting membrane (ILM), nerve fiber layer, ganglion cell (GC) layer, inner plexiform layer, inner nuclear layer, outer plexiform layer, outer nuclear layer, external limiting membrane, and outer and inner segments of PRs with 120 million rods and about 4.6 million cones [15]. PR bodies are located in the outer nuclear layer, while the inner nuclear layer is characterized by the presence of bipolar, horizontal and amacrine cells.

GCs synapse with bipolar cells in the inner plexiform layer; the visual signal is transmitted from the PR cells, the first order neurons, to second order neurons, represented by primarily bipolar and horizontal cells, and finally, to GCs, the third-order neurons. The nerve fiber layer, which consists of the GC axons, transports the visual signal through the optic nerve to the visual cortex [16].

The RPE, a monolayer of pigmented cuboid cells, is located under the PR layer, and its function is to favor the passage of nourishment from the choroid to the external neuroretinal layer and to remove waste elements deriving from PR metabolism [15]. Below the RPE are Bruch’s membrane (BrM) and the choriocapillaris: the latter can be defined as a tissue characterized by highly fenestrated capillaries originating from the posterior ciliary artery. Sympathetic nerves regulate the choriocapillaris, while intra-retinal vessels are under the control of metabolic demand in neurons [16].

Retinal tissue receives oxygen through the internal and external vascular systems; the blood supply for the internal retina comes from three vascular plexuses originating from the ophthalmic artery, a branch of the internal carotid, while choroidal vascularization delivers oxygen to the outer retina [17]. The retina, along with the macular region, is a highly metabolically active tissue that is continuously exposed to light to enable vision. Therefore, it is characterized by the production of considerable quantities of ROS [4]. Physiologically, ROS are eliminated by a number of endogenous antioxidant systems, but sometimes excessive amounts of FRs are generated that exceed the ability of antioxidant systems to remove them [18].

Moreover, the retina is a tissue highly vulnerable to OS damage due to both its exposure to visible light and UVR and to the presence of polyunsaturated fatty acids (PUFAs) in the outer PR segment membranes. PUFAs are the main target of lipid peroxidation with simultaneous alteration of PR membranes [19].

## 4. Major Retinal Diseases and Oxidative/Nitrosative Stress

### 4.1. Age-Related Macular Degeneration (AMD)

Age-related macular degeneration (AMD) is an acquired chronic disease of the macula characterized by progressive central visual loss as a result of alterations in the PR layer, RPE, BrM and choroidal complex [20]. AMD is the third most common cause of blindness after cataract and glaucoma worldwide and typically affects 10% of people older than 65 years and more than 25% of people older than 75 years [21]. Several risk factors have been identified, among which aging, cigarette smoke, light exposure, and high fat diet are the most important ones [22]. Advanced AMD can be distinguished into two types: dry or non-neovascular AMD and wet or neovascular AMD. The first accounts for the majority of all diagnosed cases, while the wet form is responsible for the majority of cases of severe visual loss [23]. It is also possible for dry AMD to become wet and vice versa [23,24].

The first sign of dry AMD is represented by drusen, retinal deposits under the RPE. Then, geographic atrophy can occur with atrophic lesions developing in the macula. In contrast, wet AMD is characterized by a phenomenon of abnormal vascular proliferation and the subsequent development of choroidal neovascular membranes (CNV), often leading to exudation and hemorrhage in the retina (Figure 1 and Figure 2). Neovascular membranes respond to anti vascular endothelial growth factor (VEGF) therapy, while this treatment is useless in dry AMD. In this case, the only available treatment today consists of a specific combination of substances according to the Age-Related Eye Disease Study (AREDS 1 and 2) for the prevention of disease progression [21,23,24,25].

Currently, OS is considered one of the main causes of AMD, and RPE cells are the primary site of OS damage [20]. RPE cells perform various functions: light absorption, transport of nutrients, and phagocytosis of shed PR outer segments. Under certain conditions of high metabolic stress, RPE cells have to increase the phagocytosis of shed PR outer segments, and this leads to increased ROS [26]. As a consequence, lipofuscin, which is a residue of lysosomal shed PR outer segment degradation, tends to accumulate under BrM, leading to RPE cell dysfunction. An additional source of ROS is represented by NADPH oxidase and peroxidase active in the phagocytic bodies where they are involved in the oxidation of fatty acids present in shed PR outer segments [27]. The amount of ROS leads to RPE degeneration, creating a vicious circle with further increase in OS [28,29,30]. Increased levels of OS markers such as MDA and 8-OHdG were found in blood serum from AMD subjects compared to non-AMD subjects, thus suggesting that there is a strict link between OS and AMD [20]. Concurrently, a higher level of carboxyethylpyrrole, a lipid peroxidation product obtained from docosahexaenoic acid under OS, has been found in the BrM of AMD eyes [20].

Zhang et al. specifically focused their attention on the role of autophagy in AMD, which has been identified as one of the main regulatory systems of OS in AMD. RPE and PR homeostasis are ensured by autophagy, a protein degradation mechanism that is damaged in RPE cells from AMD eyes. In case of prolonged exposure to high levels of ROS, AMD RPE cells lose their ability to increase SOD expression with a further increase in OS [23,31]. As a consequence, the general trend is to enhance autophagic activity in order to prevent damage to RPE and PR cells. Specifically, the crosstalk among a series of factors such as the nuclear factor erythroid 2-related factor 2 (NFE2L2), p62, peroxisome proliferator-activated receptor gamma coactivator-1 (PGC-1) and phosphatidylinositol 3-kinase/protein kinase B/mammalian target of rapamycin (PI3K/Akt/mTOR) pathways has been extensively studied. Targeting these molecules could lead to the result of increasing autophagy with a consequent decrease in OS levels [23]. NFE2L2, a transcription factor that shows protective effects against ROS in RPE cells, is able to regulate both DNA replication and transcription and mitochondrial function by binding to antioxidant response elements. In a condition of OS, once cytosolic NFE2L2 is phosphorylated, it is able to translocate to the nucleus where it induces the expression of autophagy-related genes through antioxidant response elements [23].

The NFE2L2 pathway can be activated by p62, also known as sequestosome 1, which is a scaffolding protein. After its phosphorylation, p62 binds to LC3 (ubiquitin, microtubule-associated protein, 1 light chain 3, considered to be a primary biochemical marker for autophagy activation), leading to the degradation of malfunctioning mitochondria by autophagy [32]. Moreover, p62 is also able to activate mTORC1, one of the two major protein complexes of the mTOR pathway, thereby suppressing the autophagy process.

Of particular interest is the interaction between the PGC-1 pathway, consisting of PGC-1α, PGC-1β and PGC-1-related coactivator, and NFE2L2, which seems to be a positive regulator of PGC-1, representing a cell antioxidant defense system [23].

Many molecular mechanisms are involved in AMD, thus showing the potential therapeutic role of autophagy and NFE2L2 activation. In addition, AMD recognizes a multifactorial pathogenesis that requires a better understanding to develop new therapeutical strategies. To date, the role of NO in the onset of AMD has not been fully understood. The current hypothesis is that increased OS may induce a decrease in NO levels due to peroxynitrite (ONOO) formation. According to this hypothesis, lower NO concentrations were found in the plasma obtained from AMD patients compared to controls [33,34]. A reduced effect of NO on ocular flow regulation could induce a further decrease in NO levels by eNOS involvement [35].

### 4.2. Diabetic Retinopathy (DR)

DR, one of the main leading causes of blindness worldwide [18], represents the most serious ocular complication of diabetes mellitus (DM). DM is commonly classified into type 1 diabetes mellitus, type 2 diabetes mellitus, gestational diabetes mellitus, and other forms. As a complex and multifactorial syndrome, DM is characterized by numerous symptoms, signs and complications [36]. Along with diabetic neuropathy and nephropathy, DR can be defined as a microangiopathy that occurs as a result of prolonged diabetes [37,38]. Several intra- and extravascular factors are involved in the pathogenesis of DR, with subsequent capillary closure and decreased blood flow and oxygen to the retina. The hypoxia and resulting retinal ischemia lead to the release of pro-angiogenic factors such as VEGF with consequent neovascularization [39], a process called angiogenesis, to ensure better oxygenation in the retina [38].

According to previous research studies, five classes of DR are recognized:-Class 1: identifies a normal eye, without signs of DR.-Class 2: corresponds to a form of “mild DR” and is characterized by the presence of at least one microaneurysm in all four fundus quadrants.-Class 3: called “moderate DR”, and in this phase, diabetic macular edema can occur.-Class 4: “severe DR”, in which prolonged retinal ischemia stimulates new blood vessel growth.-Class 5: also called “proliferative DR”, this form is characterized by the proliferation of new blood vessels and the appearance of scar tissues, which can result in retinal detachment [40] (Figure 3).

Usually, DR is a symmetrical condition, but a form of asymmetric DR has been identified in about 5–10% of cases. In this condition, one eye shows proliferative DR, while a form of non-proliferative, pre-proliferative or no DR is observed in the other eye for at least 2 years [37]. As discussed previously, several factors are at play in DR: neurodegeneration, involvement of MGC, inflammation and, last but not least, OS [17]. As regards OS, its involvement as one of the main pathogenetic factors of DR is well known.

The abnormality of four metabolic pathways is closely related to the hyperglycemia-mediated development of DR: (1) polyol pathway, (2) hexosamine pathway, (3) protein kinase C (PKC) pathway and (4) intracellular formation of advanced glycation end-products (AGEs) [41].

(1)Polyol pathway: in this case, glucose is first converted into sorbitol and, secondly, sorbitol dehydrogenase oxidizes sorbitol to fructose. This second reaction determines a consumption of NAD+ with a shift of the ratio NADH/NAD+ in favor of NADH oxidase and is able to use NADH to generate ROS in retinal cells, thus increasing OS [42]. Additionally, sorbitol, due to its inability to diffuse through lipid membranes, retains water with an increase in osmotic pressure and subsequent cell death, while fructose, once transformed into fructose-3-phosphate through phosphorylation and then into 3-deoxyglucosone, represents a substrate in the AGE pathway through glycosylation [43].(2)Hexosamine pathway: once glucose is converted into fructose-6-phosphate by phosphorylation, it is subsequently transformed into glucosamine 6-phosphate by fructose-6-phosphate aminotransferase; this is the substrate of a series of enzymes involved in acetylation and isomerization reactions that first form N-acetylglucosamine 6-phosphate and then diphosphate uracil-N-acetylglucosamine, the end product of hexosamine pathway. High activation of this metabolic pathway provides ROS with overproduction in mitochondria and mitochondrial respiration alteration, thus further increasing the amount of ROS [44].(3)PKC pathway: hyperglycemia is able to stimulate the glycolysis pathway and, as a consequence, the amount of diacylglycerol. In case of increased diacylglycerol, PKC, a family of 12 serine/threonine kinases, is highly activated. The PKC pathway is involved in regulating several processes such as endothelial permeability, expression of VEGF and leukostasis in the retinal cells. Furthermore, PKC can activate NADPH oxidase, thus inducing the production of ROS [44,45].(4)AGEs: excess glucose leads to the endogenous nonenzymatic glycosylation of macromolecules, which induces the synthesis of AGEs [46]. There are also exogenous sources of AGEs, such as cigarette smoke and certain foods or drinks that contain these products [47]. Pentosidine, carboxymethyllysine and carboxyethyllysine represent the most studied AGEs in humans [48]. AGEs induce irreversible retinal damage by binding to specific receptors called receptor for advanced glycation end products (RAGEs). They cause retinal vessel occlusion resulting in an ischemic insult, besides activating a series of intracellular signaling pathways, such as mitogen activated protein kinase/nuclear factor kappa-light-chain-enhancer of activated B cells (MAPK/NF-κB). A cycle of hypoxia, cytosolic ROS augmentation and antioxidant defense consumption is established with subsequent activation of the inflammation cascade [44].

In addition to this process, α-ketoaldehyde methylglyoxal, which is a low-molecular-weight AGE precursor, is overproduced by a series of enzymes such as triose phosphates, dihydroxyacetone phosphate and glyceraldehyde-3-phosphate. The α-ketoaldehyde methylglyoxal is highly cytotoxic and is able to react with several macromolecules, damaging them and stimulating pericyte apoptosis [48].

A phenomenon worth bearing in mind is “metabolic memory”. This is a severe dysfunction that can be explained as a sort of vasculature cell imprinting responsible for alterations in microvasculature even in the presence of glycemic control.

This particular condition helps to shed light on DR development in DM patients under strict glycemic control. Some authors consider “metabolic memory” a consequence of gene expression and mitochondrial function alteration occurring in the early phases of the diseases [49].

In addition to dysfunction of these four pathways in DR, the role of MGC in retinal antioxidant defense is worth mentioning. MGCs were discovered by Heinrich Müller in the mid-19th century and at first, they were thought to be support cells without a specific function. Later, MGCs were identified as crucial cells in the anatomy and physiology of the retina, being able to interact with cells in all retinal layers and to play an important role in DR [17].

In conditions of hyperglycemia, heme oxygenase-1 expression and increased ROS production in MGCs is observed [50,51,52]. Normally, MGCs respond to increased OS by producing glutathione (GSH), one of the most important cell antioxidant molecules. GSH stimulates GABA release to prevent excitotoxicity in the retina [53]. A reduction in GSH levels has been observed in the rat retina after 3 weeks of hyperglycemia, probably causing OS in DR [54]. In addition to OS, also worth bearing in mind is the contribution of nitrogen radicals in the pathogenesis of DR: in hyperglycemia, superoxide reacts with NO, producing ONOO, which is able to induce cell damage [10].

Of particular interest is the study conducted in 2010 by Li et al., who observed that eNOS knockout (−/−) mice, in which diabetes was induced, showed greater retinal vascular permeability than controls due to an increase in NO concentration, which they attributed to the high expression of iNOS [55].

Other studies have highlighted a close link between NO plasma concentrations and the severity of DR: the more severe the DR, the higher the NO plasma levels [56]. All of these discoveries focus their attention on the role of OS and NO in DR and, specifically, shed light on MGC and OS/NO as potential therapeutic targets of DR.

### 4.3. Retinitis Pigmentosa

The term “retinitis pigmentosa” (RP) refers to a set of complex hereditary retinal dystrophies affecting 1.5 million people worldwide [57]. This condition is due to progressive PRs and RPE degeneration leading to specific visual symptoms [58]. Usually, the rods are affected first, with subsequent involvement of cones with typical night blindness followed by peripheral visual field deterioration in daylight [Figure 4 and Figure 5]. Two different forms of RP have been described; the most common is non-syndromic RP, but many syndromic forms exist, such as Usher syndrome [59].

Both genetic and non-genetic factors contribute to RP pathogenesis. RP is an extremely heterogenous disease, and every kind of transmission pattern has been reported in the last few years. Nearly 50 genes have been related to non-syndromic RP, and 3100 mutations have been identified in these genes. As regards syndromic RP, 12 genes are known to be linked to Usher syndrome, and 17 genes are associated with Bardet–Biedl syndrome. As regards non-genetic factors, the most critical are inflammation, immune response, vascularization and OS [60].

Recently, the central role of OS in the pathogenesis of RP has been underlined by Gallenga et al., who observed that several mutations in endogenous antioxidant pathways such as mutY DNA glycosylase (MUTYH), ceramide-kinase like (CERKL) and glyoxalase 1 (GLO1) are frequently implicated [61]. For instance, MUTYH contributes to regulating genomic integrity through the base excision repair pathway [62]. In case of DNA damage by OS, MUTYH is over-activated, leading to single-strand breaks of DNA, followed by cell death. Oka et al. highlighted the crucial role of MUTYH-mediated base excision repair in RP retinal alterations. In an rd 10 mouse model, they observed a decrease in 8-oxoguanine, the product of DNA oxidation, determined by MUTYH, and as a consequence, rod and cone death [63]. High levels of 8-oxoguanine and other oxidized DNA are usually present both in the PR layer and in microglial cells and macrophages in the outer retinal regions. Therefore, these inflammatory cells could represent a source of ROS and a possible therapeutic target in RP [62,63,64]. The activation of different microglial phenotypes and their differentiation, dependent on the environment (M1, M2 and intermediate) in the development of RP, is particularly compelling. As reported by several authors in an rd1 mouse line, microglial markers such as tumor necrosis factor, interleukin 1α, interleukin 1β, transmembrane protein 119 and myeloid cells are over-expressed in the early phases of the disease where they mediate neurotoxicity [65,66,67].

However, microglia cells have controversial functions, involving both neurotoxic and neuroprotective/neurotrophic effects; therefore, their role still remains to be clarified.

As regards CERKL, this is a gene encoding for a molecule implicated in stress-induced apoptosis. The increase in ceramide can activate an apoptosis pathway. To avoid this situation, phosphorylation of ceramide-by-ceramide kinase is induced. Recently, different types of RP characterized by specific macular and peripheral lesions have been closely related to CERKL mutations [68,69].

Another enzyme whose mutations have been identified in RP is GLO1: it prevents the accumulation of the cytotoxic products of glycolysis, which are the main source of AGE formation. Growing evidence suggests the role of GLO1 mutations in RP pathogenesis [70,71,72].

As regards the role of nitrosative stress, Benlloch-Navarro et al. observed a decrease in iNOS levels in the retina of rd10 mice, a type of RP animal model, while no changes in nNOS concentration were revealed [73].

Kanan et al., starting from the evidence that nitrosative stress is increased in RP models, administered rd10 mice daily subcutaneous injections of 40 mg/kg of metipranolol, a β-adrenergic receptor antagonist used in the treatment of hypertension, and observed a reduction in both PR loss and in markers of nitrosative stress in rd10 mice, thus suggesting a possible therapeutic role for this drug in RP [74].

### 4.4. Stargardt Disease

Stargardt disease (STGD1) is the most common inherited macular dystrophy, autosomal recessive, with a prevalence of 1:8000–10,000 [75]. Typically occurring in childhood or early adolescence, STGD1 is due to mutations in ABCA4 (ATP-binding cassette, sub-family A, member 4; OMIM #601691), a gene first identified in 1997 [76] and encoding “flippase”, a retinal transporter protein located in the rim of PR disks whose function is to eliminate retinaldehyde, a toxic product for retina [77,78].

The ocular fundus aspect is characteristic: patients affected by STGD1 show macular atrophy along with RPE typical yellow-white flecks [79]. Symptoms referred by these patients are represented by progressive and bilateral central vision loss; in case of childhood-onset STGD1, symptoms and clinical features can be worse than the ones reported by subjects affected by adult-onset or the later onset "foveal-sparing" (FS) STGD1 forms.

It has long been known that there is a close link between the age of onset and the severity of the disease [75,79]. Additionally, in the case of STGD1 pathogenesis, OS seems to be involved, as Taubits et al. highlighted in some Stargardt disease murine models.

Studying albino Abca4−/− and Abca4−∕−.Rdh8−∕− mice, they found some OS dysfunctions that were absent in pigmented Abca4−∕− mice, thus suggesting the protective role of melanin against OS.

According to this finding, they used immunochemistry to evaluate the presence of OS markers and they found the products of lipid peroxidation, in particular 4–hydroxynonenal, malondialdehyde and other molecules, in the RPE cells of Abca4−∕− and Abca4−∕−.Rdh8−∕− mice [78]. They concluded that the presence of melanin in RPE is a crucial factor in retinal health that helps in preventing early RPE degeneration, both in AMD and Stargardt disease [78].

### 4.5. Retinopathy of Prematurity

First described in 1942 by T. L. Terry as retrolental fibroplasia [80], retinopathy of prematurity (ROP) is a multi-factorial pathology of retinal development and typically occurs in premature babies, involving the retinal vascular network [81].

Despite several improvements in therapeutic strategies, ROP still represents the second most common cause of childhood blindness [82]. ROP usually develops in premature babies who are exposed to high oxygen flows, and this is the cause of FR generation [83]. Gestational age and birth weight have long been known to represent the two greatest risk factors for ROP, which feature an inverse correlation: younger children and those born at an earlier gestational age are more at risk of developing ROP [80].

The clinical presentation of ROP is highly heterogenous: sometimes a spontaneous regression is observed, sometimes the situation becomes complicated with bilateral retinal detachment, leading to total blindness [82]. Preterm infants can be characterized by incomplete retinal vascularization at birth, presenting some avascular areas capable of promoting VEGF production [80]. Two distinct phases of ROP can be recognized: at first, an ischemic phase is observed due to the interruption of the progression of retinal capillary genesis; then, a proliferative phase follows, being characterized by a phenomenon of neoangiogenesis under the stimulus of VEGF released by ischemic areas [82].

Among the several factors involved in the genesis of ROP, OS plays a significant role. At birth, the newborn undergoes a fast change in oxygen levels, due to the passage from a low to a higher oxygen concentration environment [84,85]. Both hyperoxia and hypoxia are directly involved; along with these factors, nitro-OS tends to increase under conditions of OS [86]. Nitrite, nitrate and ONOO are formed with subsequent retinal microvascular injury [87].

The mechanisms by which NO-derived reactive species determine microvascular damage are not yet well characterized. Recently, an NO_2_-mediated peroxidation process was elucidated. The attention is focused on some nitrative stress mediators and specifically on the products of NO_2_-mediated arachidonic acid isomerization trans-arachidonic acid isomers. Trans-arachidonic acid isomers participate in microvascular injury by inducing an upregulation of thrombospondin-1 in retinal ECs. Thrombospondin-1 is able to inhibit angiogenesis by inducing microvascular EC apoptosis. In preterm infants, this process may lead to the development of ROP [88].

Alongside trans-arachidonic acid, in recent years other biomarkers of OS have been investigated as possible targets in order to prevent oxidative damage in preterm infants. Notably, when comparing groups of patients affected by ROP to groups of unaffected patients, some markers were highly or less expressed in the first group:-Malondialdehyde, one of the molecules of lipid peroxidation and a sensitive OS marker.-8-hydroxy-2-deoxyguanosine, a biomarker of oxidative DNA damage.-Levels of GSH, a nonenzymatic antioxidant of cytoplasm, which was lower in ROP patients [88].

In relation with all of these efforts to identify increased OS molecules in this type of patient, the future prospects are to develop increasingly specific novel therapeutic strategies for preterm infants at risk of ROP.

### 4.6. Eales’ Disease (ED)

Eales’ disease (ED) is commonly considered an idiopathic vasculitis that mainly involves the peripheral retina [89]. First described by Henry Eales in 1880 [90], ED typically arises as a monolateral peripheral perivasculitis followed by both capillary non-perfusion and proliferative manifestations, leading to recurrent vitreous hemorrhage [91]. However, in 50–90% of patients, bilateral involvement is observed [89]. Several complications of ED such as cataract, neovascular glaucoma, rubeosis iridis, tractional retinal detachment and bulbar phthisis have been described [89].

It is quite rare in the West, being more frequent in Asia and especially in the Indian subcontinent where it is frequently observed in young men between 20 and 40 years of age [89]. ED pathogenesis has not yet been clarified, but it is common opinion that ED is a form of hypersensitivity to tuberculin-protein occurring after exposure to Mycobacterium tuberculosis [92]. In the last few years, the role of human leucocyte antigen, retinal S antigen autoimmunity, hyperhomocysteinemia, and, specifically, OS have been taken into consideration in its etiopathogenesis [93].

More in detail, inflammation plays a crucial role in the pathogenesis of ED: as a result of an increased inflammatory stimulus, the blood retinal barrier shows increased permeability, resulting in the passage of inflammatory mediators, such as lymphocytes and monocytes, into retinal layers. These cells respond to inflammation by producing ROS and RNS. In this regard, one of the main mediators of inflammation and ischemia/reperfusion injury is ONOO [94,95].

The latter molecule is generated by the reaction between the superoxide anion and NO. ONOO is a highly reactive substance. Considering the diminished SOD activity observed in mitochondria from ED patients, Hallywell et al. postulated the involvement of RNS in SOD alteration and related decreased levels [94]. Reduced SOD levels lead to an increase in superoxide anion, thus resulting in release of iron (Fe) and Cu from enzymes involved in the respiratory chain [96,97]. High levels of Fe and Cu in ED mitochondria have been observed and correlated with high lipid peroxide levels in these patients [94].

Further evidence of the involvement of OS in ED is represented by the finding of increased levels of thiobarbituric acid reacting substances and decreased levels of SOD and GSH in vitreous samples obtained from these patients [98].

### 4.7. Retinal Neurodegeneration

Neurodegenerative eye diseases can be responsible for severe low vision or legal blindness and are characterized by progressive degeneration of retinal GCs. The subsequent axonal damage will lead to neurodegeneration of the optic nerve with loss of the visual field, central vision, and sensitivity to color and contrast both photopic and scotopic. The sensory deficit will be more or less severe depending on the neurodegenerative damage that will occur [99].

In particular, glaucoma is characterized by progressive degeneration of retinal GCs, and one of the risk factors for this disease is increased intraocular pressure. However, some patients do not respond to therapy with reduction of intraocular pressure. In fact, in normal tension glaucoma, the disease advances with progressive degeneration of retinal GCs. Some researchers argue that another important risk factor in human glaucoma is OS. They evaluated the level of GSH in plasma, one of the main antioxidants, and found that it was reduced in these patients [99,100].

Furthermore, vascular endothelial dysfunction has been implicated in a primary role in the development of neurodegenerative diseases in the retina by means of OS [101]. ROS play a role significant to the degeneration of neuronal cells by modifying the function of biomolecules. The vascular ECs regulate vascular permeability, send out messengers for the biological functions, and are continuously subjected to hemodynamic forces. Moreover, ECs secrete anticoagulants, procoagulants, and fibrinolytic and angiogenic growth factors, and regulate the expression of cytokines, chemokines, and adhesion molecules. In the retina, ECs interact with the neurovascular cells to regulate the blood–retina barrier, neuronal metabolism, and immune system. Like this, once the transmission is activated, the intracellular reactions favor the release of vasodilators and vasoconstrictors such as NO, prostaglandins, endothelin, thromboxane A2, and endothelium-derived relaxing factor [99,100].

It is well known that the endothelial dysfunction is impacted by increased OS. The vascular endothelial dysfunction impairs ocular hemodynamics by reducing the bioavailability of NO and increasing the production of ROS. Moreover, we know that amyloid β is produced from amyloid precursor protein by the actions of β- and γ-secretases, and amyloid β is observed in more than 80% of Alzheimer disease patients as the deposition of amyloid in the parenchyma and cerebral vessels. This condition is called cerebral amyloid angiopathy, but the mechanism for how amyloid-β accumulates in blood vessels remains largely unknown [99,100,101].

Therefore, the activation of inflammatory cytokines, the expression of adhesion molecules on the surface of vascular cells, decreased levels of NO, insufficient blood supply, and the accumulation of amyloid-β peptide lead to neurodegenerative changes in the brain and in the eyes, and are closely associated with Alzheimer’s disease and glaucoma [99,100,101].

Consequentially, it is recognized that EC dysfunction is crucial in the development of neurodegenerative diseases as a primary target of excessive glucocorticoid and catecholamine action [99].

## 5. Discussion and Conclusions

Today, there is increasing interest in the involvement of oxidative and nitrosative stress in a whole series of retinal diseases, including macular diseases. Many advances are being made in understanding the molecular mechanisms involved in the various pathologies; however, concrete therapies are still lacking. The hope is that, by continuing research, it will be possible to develop increasingly personalized therapies based on genetics. It is, therefore, highly probable that oxidative and nitrosative stress will be the starting points for most of the therapies in the coming years.

We know several factors contribute to the etiopathogenesis of ocular disorders, but OS seems to play a prominent role in the various representative forms of retinopathies of degenerative, genetic, senile, inflammatory, and dysmetabolic origin. We wanted to briefly summarize what is reported in the international literature on this topic. In fact, although OS is a source of heated discussion and research in the scientific field, there are still numerous study models and paths that must be pursued in order to resolve the questions surrounding the various pathologies.

With our review, we wanted to sensitize researchers in the ophthalmology field, as the eye represents a privileged organ thanks to its anatomical and functional autonomy that could allow an easier scientific approach to research.

Indeed, the ocular district, in its context, allows a direct study of the etiopathogenetic causes, but also a possible direct clinical/surgical therapeutic solution. In this regard, the use of antioxidant molecules, stem cells, induced pluripotent stem cells and gene therapy in association with inactive adenoviruses or lipid or synthetic polymers would represent a valid example.

Unfortunately, a limitation of our work is the fact that it is not possible to evaluate in detail all of the articles that have been published on the subject over the years. Therefore, we mainly cited those works that we believe should be taken seriously by researchers, leaving them discretion.

In summary, we hope that further scientific study in this area will favor the clarification of other systemic diseases and consequently the possibility of providing appropriate therapies for specific retinal diseases.

## Figures and Tables

**Figure 1 antioxidants-11-01790-f001:**
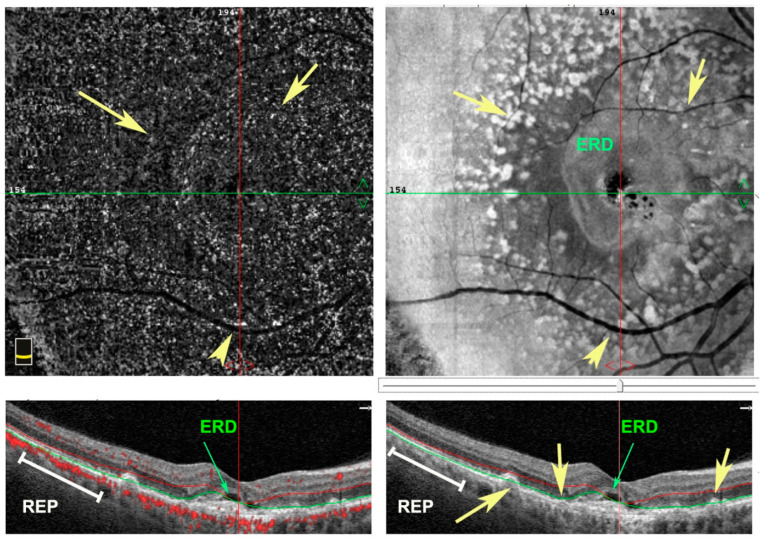
En face (**upper** panel) and structural B-scan (**bottom** panel) images of spectral domain-optical coherence tomography (SD-OCT) of a right eye. Patient affected by wet age-related macular degeneration (AMD) with exudative retinal detachment (ERD) in the foveal area. Numerous drusen deposits as hyperfluorescent dots (yellow arrows) are under retinal pigment epithelium (RPE) (white lines).

**Figure 2 antioxidants-11-01790-f002:**
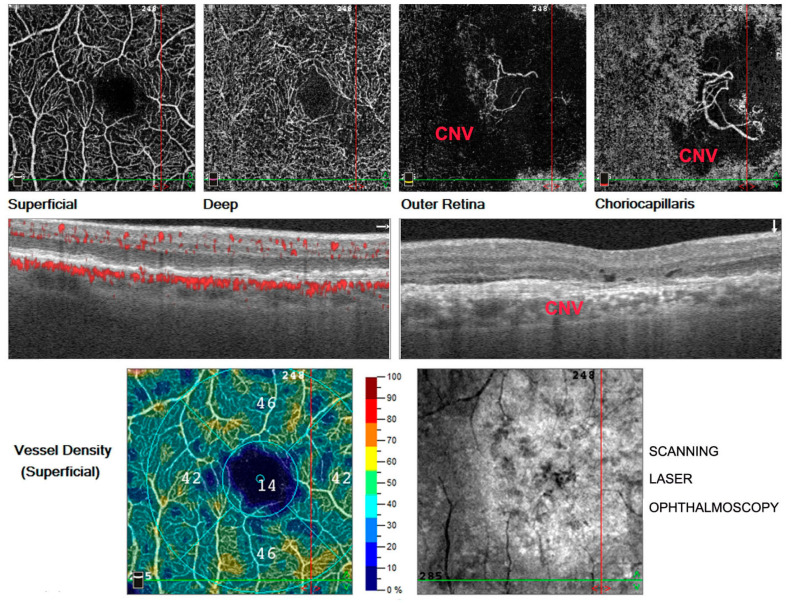
Optical coherence tomography angiography (OCTA) in right eye. Images of a patient suffering from wet age-related macular degeneration (AMD) characterized by a phenomenon of abnormal vascular proliferation in the choriocapillaris layer and consecutive choroidal neovascular membrane (CNV) development in the macular region, clearly evident in the last image at the top right.

**Figure 3 antioxidants-11-01790-f003:**
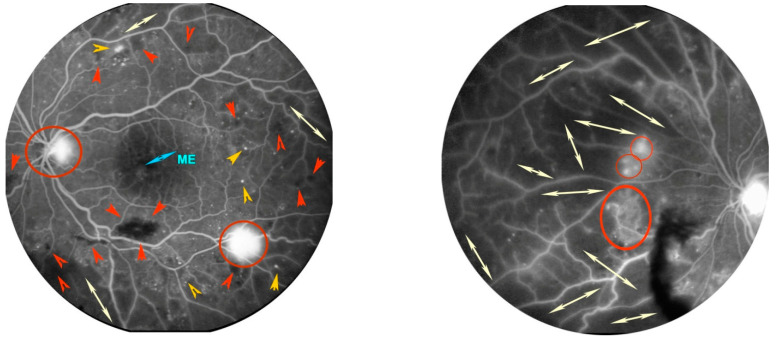
Fluorescein angiography of a patient with proliferative diabetic retinopathy (DR) in left eye. Posterior pole in the left image. Presence of numerous microaneurysms as hyperfluorescent dots (orange arrow heads), bleeding as irregular black spots (red arrow heads), and initial macular edema (ME) as retinal foveal hyperfluorescence. Retinal nasal quadrant in the right image. Presence of large dark areas without terminal vessels due to retinal ischemia (white arrows). In both images, some hyperfluorescent spots on the optic disc and along the nasal and temporal vascular arches are highlighted, due to proliferation of new blood vessels (red circles).

**Figure 4 antioxidants-11-01790-f004:**
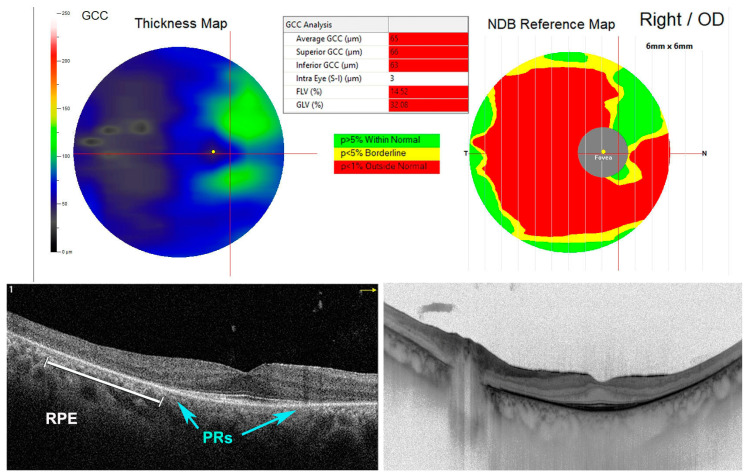
Spectral domain-optical coherence tomography (SD-OCT) of a patient with retinitis pigmentosa (RP). In right eye, it is possible to notice the absence of photoreceptors (PRs) (blue arrows) and a retinal pigment epithelium (RPE) degeneration (white line) beyond the perifoveal area. Thinned ganglion cell complex (GCC thickness map). Outside normal retinal thickness distribution was established by the normative database reference map (NDB reference map).

**Figure 5 antioxidants-11-01790-f005:**
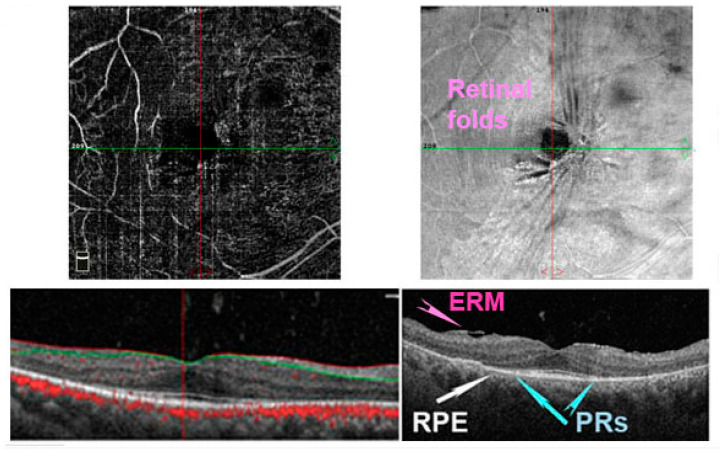
Optical coherence tomography angiography (OCTA) in a patient with retinitis pigmentosa (RP). In right eye, it is possible to notice the absence of photoreceptors (PRs) (blue arrows) and a retinal pigment epithelium (RPE) degeneration (white arrows) beyond the perifoveal area. The vascular plexus is reduced (upper left image). Moreover, an epiretinal membrane (ERM) is detected in the enface image (upper right image) and structural image (bottom right panel) forming some retinal folds.

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
