# Peer review of "Oxidative Stress Implication in Retinal Diseases—A Review"

_antioxidants, 2022, doi:10.3390/antiox11091790_

Round 1

Reviewer 1 Report (Previous Reviewer 1)

Thank you for addressing all of my comments. I recommend accepting this manuscript. 

Reviewer 2 Report (Previous Reviewer 2)

No other comments.

This manuscript is a resubmission of an earlier submission. The following is a list of the peer review reports and author responses from that submission.

Round 1

Reviewer 1 Report

The authors describe an important, relevant topic, which presents one of the major unmet clinical needs of today. Nevertheless, I have a number of concerns and comments regarding the readability and content of this manuscript. 

The sentence in lines 24-25 seems redundant for the abstract, since it is already mentioned in line 44 and 45, unless authors tend to focus mainly on RNS: "In particular, scientific literature focuses more on 24 ROS participation in the pathogenesis of diseases than on the role played by RNS." 

What is the reason for starting a new paragraph after each sentence? 

The authors are highlighting the lack of discussion on the reactive nitrogen species in both abstract and introduction, as I mentioned earlier. Why is that, and why don't the authors explore this a bit further, since it is an important topic, as per their description?

Line 69: which pathway?

For oxygen molecules, please put 2 as a subscript. 

Line 86, "which" would be added, or the sentence should be separated in 2. 

Line 111 "sveral"

Suggestion for the authors would be making an illustration depicting the complex histology of the retina. 

Arrows should be added to figures for better understanding by the reader. Also, I believe these are screenshots, is every part of the picture useful for reader's understanding of the content (the authors only explain the top right pictures in most figures)? 

Line 385-385. The authors can list different microglial phenotypes, and their differentiation dependent on the environment( M1, M2 and intermediate).

A more detailed discussion is needed on this topic. What is the value of this work, and what are the limitations? What are the future directions?

Thank you. 

Reviewer 2 Report

The authors reviewed contribution of oxidative stress to retinal diseases.

The main problem is that there are already multiple reviews relevant for this topic. Some examples are provided below:

Inherited Retinal Dystrophies: Role of Oxidative Stress and Inflammation in Their Physiopathology and Therapeutic Implications. Pinilla I, Maneu V, Campello L, Fernández-Sánchez L, Martínez-Gil N, Kutsyr O, Sánchez-Sáez X, Sánchez-Castillo C, Lax P, Cuenca N. Antioxidants (Basel). 2022 May 30;11(6):1086. doi: 10.3390/antiox11061086.

The Intertwined Roles of Oxidative Stress and Endoplasmic Reticulum Stress in Glaucoma. Hurley DJ, Normile C, Irnaten M, O'Brien C. Antioxidants (Basel). 2022 Apr 29;11(5):886. doi: 10.3390/antiox11050886.

Oxidative Stress, Vascular Endothelium, and the Pathology of Neurodegeneration in Retina. Shi X, Li P, Liu H, Prokosch V. Antioxidants (Basel). 2022 Mar 12;11(3):543. doi: 10.3390/antiox11030543.

The Impact of Oxidative Stress on Blood-Retinal Barrier Physiology in Age-Related Macular Degeneration. Tisi A, Feligioni M, Passacantando M, Ciancaglini M, Maccarone R. Cells. 2021 Jan 4;10(1):64. doi: 10.3390/cells10010064.

The Relevance of Oxidative Stress in the Pathogenesis and Therapy of Retinal Dystrophies Elena B. Domènech and Gemma Marfany Antioxidants (Basel). 2020 Apr; 9(4): 347.  doi: 10.3390/antiox9040347

Oxidative stress and diabetic retinopathy: Molecular mechanisms, pathogenetic role and therapeutic implications. Kang Q, Yang C. Redox Biol. 2020 Oct;37:101799. doi: 10.1016/j.redox.2020.101799.

Oxidative stress in the light-exposed retina and its implication in age-related macular degeneration. Ozawa Y. Redox Biol. 2020 Oct;37:101779. doi: 10.1016/j.redox.2020.101779.

Oxidative Stress and Microglial Response in Retinitis Pigmentosa. Murakami Y, Nakabeppu Y, Sonoda KH. Int J Mol Sci. 2020 Sep 28;21(19):7170. doi: 10.3390/ijms21197170.

Autophagy in Age-Related Macular Degeneration: A Regulatory Mechanism of Oxidative Stress. Zhang ZY, Bao XL, Cong YY, Fan B, Li GY. Oxid Med Cell Longev. 2020 Aug 8;2020:2896036. doi: 10.1155/2020/2896036.

Mechanisms of mitochondrial dysfunction and their impact on age-related macular degeneration. Kaarniranta K, Uusitalo H, Blasiak J, Felszeghy S, Kannan R, Kauppinen A, Salminen A, Sinha D, Ferrington D. Prog Retin Eye Res. 2020 Nov;79:100858. doi: 10.1016/j.preteyeres.2020.100858.

 Oxidative Stress and Microvascular Alterations in Diabetic Retinopathy: Future Therapies. Rodríguez ML, Pérez S, Mena-Mollá S, Desco MC, Ortega ÁL. Oxid Med Cell Longev. 2019 Nov 11;2019:4940825. doi: 10.1155/2019/4940825.

 Oxidative Stress, Ocular Disease and Diabetes Retinopathy. Tangvarasittichai O, Tangvarasittichai S. Curr Pharm Des. 2018;24(40):4726-4741. doi: 10.2174/1381612825666190115121531.

The Oxidative Stress and Mitochondrial Dysfunction during the Pathogenesis of Diabetic Retinopathy. Wu MY, Yiang GT, Lai TT, Li CJ. Oxid Med Cell Longev. 2018 Sep 5;2018:3420187. doi: 10.1155/2018/3420187.

Ischemic Retinopathies: Oxidative Stress and Inflammation. Rivera JC, Dabouz R, Noueihed B, Omri S, Tahiri H, Chemtob S. Oxid Med Cell Longev. 2017;2017:3940241. doi: 10.1155/2017/3940241.

Diabetic retinopathy: hyperglycaemia, oxidative stress and beyond. Hammes HP. Diabetologia. 2018 Jan;61(1):29-38. doi: 10.1007/s00125-017-4435-8.

The impact of oxidative stress and inflammation on RPE degeneration in non-neovascular AMD. Datta S, Cano M, Ebrahimi K, Wang L, Handa JT. Prog Retin Eye Res. 2017 Sep;60:201-218. doi: 10.1016/j.preteyeres.2017.03.002.

Targeting Oxidative Stress for Treatment of Glaucoma and Optic Neuritis. Kimura A, Namekata K, Guo X, Noro T, Harada C, Harada T. Oxid Med Cell Longev. 2017;2017:2817252. doi: 10.1155/2017/2817252.

 Oxidative Stress-Related Mechanisms and Antioxidant Therapy in Diabetic Retinopathy. Li C, Miao X, Li F, Wang S, Liu Q, Wang Y, Sun J. Oxid Med Cell Longev. 2017;2017:9702820. doi: 10.1155/2017/9702820.

Retinal Diseases Associated with Oxidative Stress and the Effects of a Free Radical Scavenger (Edaravone). Masuda T, Shimazawa M, Hara H. Oxid Med Cell Longev. 2017;2017:9208489. doi: 10.1155/2017/9208489.

Oxidative Stress in Retinal Diseases Yuhei Nishimura, Hideaki Hara, Mineo Kondo, Samin Hong, Takeshi Matsugi Oxid Med Cell Longev 2017;2017:4076518. doi: 10.1155/2017/4076518

1) The search of the PubMed database reveals 218 review papers published from the beginning of the 2020 that are relevant for the topic: oxidative stress retinal disease. Please explain why we need one more review about the oxidative stress retinal diseases. Explain what is the new insight provided by the submitted paper.

Other issues.

2) The paper requires extensive English editing.

3) Please, avoid acronyms. Their usage makes it difficult to read papers.  As stated recently, acronyms are not the biggest current problem in science communication, but reducing their use is a simple change that would help readers and potentially increase the value of science (doi: 10.7554/eLife.60080).  Please, look at this paper, it is interesting and fully consistent with my own experience.

4) The authors write: “….. apoptosis, which is an indispensable process for normal cellular homeostasis…..”.

 Apoptosis is a cell death and, therefore, it can not maintain cellular homeostasis.  Please, rewrite this statement. It will be OK If you will replace “cellular homeostasis” with “tissue homeostasis”.